# Non-Volatile Memory Forensic Analysis in Windows 10 IoT Core

**DOI:** 10.3390/e21121141

**Published:** 2019-11-22

**Authors:** Juan Manuel Castelo Gómez, José Roldán Gómez, Javier Carrillo Mondéjar, José Luis Martínez Martínez

**Affiliations:** Albacete Research Institute of Informatics, 02071 Albacete, Spain; jose.roldan@uclm.es (J.R.G.); javier.carrillo@uclm.es (J.C.M.); joseluis.martinez@uclm.es (J.L.M.M.)

**Keywords:** cybersecurity, forensics, IoT, Windows 10 IoT Core

## Abstract

The increase in the number of cybersecurity incidents in which internet of things (IoT) devices are involved has called for an improvement in the field of computer forensics, which needs to provide techniques in order to perform complete and efficient investigations in this new environment. With the aim of doing so, new devices and systems are being studied in order to offer guidelines for investigators on how to examine them. This papers follows this approach and presents a forensic analysis of the non-volatile memory of Windows 10 IoT Core. It details how the investigation should be performed and highlights the relevant information that can be extracted from storage. In addition, a tool for the automation of the retrieval of the pieces of evidence detected is provided.

## 1. Introduction

Among the different definitions of the term things, one of them describes it as “an object not specifically named or designated”. Even though it might not seem that this non-specific concept can be applied in a scientific context, it turns out that it is completely accurate when used to describe the new paradigm existing in computer science. The internet of things (IoT) offers such a wide range of options and features that it is not possible to narrow it down. IoT devices can be present anywhere, and we are using them without even noticing. Everyday technology users will find themselves using drones, smart TVs, smart speakers or simply sensors in their home to measure temperature. Nevertheless, the context in which IoT devices are present is not only limited to the smart home environment, as other fields such as eHealth or smart industries have their origin in the application of the IoT in certain scenarios.

According to a Gartner estimation [1], in 2018 there were more than 11 billion IoT devices installed, and an increase of almost twice this value is predicted for 2020, with 20.4 billion. Regarding the context in which they are used, the consumer segment is the one where more IoT units are installed, accounting for 63% of them. The coexistence of this huge number of devices translates into an advantage for consumers, offering a wide variety of options to choose from, and numerous contexts in which they can use IoT devices, but this is a big inconvenience for developers. The heterogeneity of the platform hinders the establishment of a common ground to be shared by all the systems.

This is not the only negative aspect of the IoT, as a greater concern involves cybersecurity. The security measures implemented on the devices, especially during the IoT’s conception, turned out to be a huge underestimation of the requirements that these devices and the information that they handle demand. The prioritization of usability, added to the failure at that time to appreciate that something as simple as a smart bulb could compromise the security of an entire network, resulted in a false sense of security. Nowadays, companies and developers have acknowledged this issue, taking steps to improve the protection of the devices and their information, although there is still a very long way to go. It must be remembered that the data that this environment is handling is very sensitive, especially in contexts such as eHealth or smart homes. In addition, IoT devices are also present in critical environments, carrying out very delicate tasks, so the impact that an incident might have in these scenarios could be catastrophic. Furthermore, this is a concern that is currently having negative consequences, so time is working against us. The need to implement proper security on these devices is imperative.

This situation has created a perfect environment for cybercriminals, as they can obtain high rewards with very little effort. This is evidenced by recent studies in which malware samples explicitly designed for the IoT are analyzed. In the year 2018, more than 120 thousand malware samples were detected, which was an increment of almost four times compared with 2017 [2], with distributed denial of service (DDoS) attacks, cryptocurrency mining and data theft being the most common types. Most worrisome is the infection vector used by them; the Mirai botnet family took control of the system through a dictionary attack on the devices that still had the default credentials [3]. It might seem an obvious attack that would have had no impact on a system, but it was just the opposite. The first version affected more than 600 thousand IoT units, and, with its different variations, went on to infect millions, proving the weakness of the security measures of the devices. Such was the success of this malware that in 2018, two years after the first version was detected, 20.9% of the samples of IoT malware belonged to the Mirai family, and new versions still appear every day. A more recent case is the malware Silex, which in June 2019, also targeting systems with default login credentials, infected thousands of devices and wiped their firmware, confirming once more that, years later, most of the security measures are still powerless against the simplest attacks [4]. Therefore, the magnitude of the problem is already significant, and it is becoming greater every year.

### 1.1. Problem Discussed and Research Motivation

The weakness of the security measures of IoT devices and systems, combined with the appearance of IoT malware samples, has translated into an increase in the number of cyberincidents. In order to respond to these incidents, and to determine what has happened in them, researchers are developing forensic techniques, adapting them to the characteristics of the environment in which the incidents take place. Given the novelty of the IoT, the current state of the art has not been adjusted to it yet. As a consequence, the only option for investigators is to follow conventional approaches and try to modify them in order to be able to carry out the examinations. This produces inefficient forensic investigations, and, what is most concerning, can even lead to the inadmissibility of evidence if it is not handled appropriately.

In order to comprehend how distinct the characteristics of the IoT environment are compared with those of conventional forensics, the most relevant ones are listed and described below. In addition, we also explain how each one of them affects the investigation process, and why it complicates the development of techniques or the use of conventional ones. These characteristics are the following:

Purpose. The main characteristic, although it may seem obvious, is the functionality for which IoT has been conceived, which is what shapes the creation of devices and systems. They have not been designed to improve the performance of previous devices, but to provide new contexts with technology or ones that did not have any. Therefore, some of the IoT systems have scarcely any similarities with conventional ones, so, in certain cases, they cannot be used as a reference.

Connectivity. It is quite common to find IoT networks in which multiple devices are present. In fact, most of the topologies are based on the interaction between the units present in it. As a consequence, a cyberincident will likely affect more than a single device and, if it is not the case, the data that has been exchanged in the network will be a valuable source of information. This increases the range of forensic investigations, as more devices have to be studied, and changes the perspective from which it has to be addressed, as now that perspective becomes a collective one.

Computational capacity. One of the main characteristics of the context is the reduction in computational capacities in exchange for mobility. Consequently, the storage of the devices has been reduced significantly compared with that of conventional ones, meaning that fewer sources of evidence are available for investigators. It also affects the lifetime of the data, which is now shorter. In addition, the lack of computational power complicates the possibility of carrying out a live analysis, since it takes more time to execute tasks. Therefore, understanding how a device or system works is more crucial than ever, so that no evidence is left out.

Location. IoT devices have been designed to be installed in confined places or even be embedded into objects. This, added to the fact that two devices in the same network can be in very distant locations, which means that the investigator may not always have physical access to them. As a consequence, the acquisition, and analysis phases must be adapted in order to provide techniques to follow a live approach, which is not very common in conventional forensics.

Heterogeneity. There are a great number of devices that coexist in the IoT, and they are used for very diverse purposes. E-health, smart cities or smart industry are three examples of different contexts in which IoT devices are present, and each one of them has its own characteristics and requirements. In fact, there are systems that have been designed to be only used in a specific scenario. Consequently, it is quite difficult for the forensic community to develop solutions, such as tools or methodologies, that can satisfy the requirements of all of them by following a general approach.

For these reasons, researchers have opted to study independent IoT devices and systems, so that their characteristics can be taken into account when designing new methodologies and tools, and also to provide guidelines on how to examine them. Given the mentioned heterogeneity of the environment, there are a great number of systems that are unknown, forensically speaking, particularly those that have been designed specifically for the IoT. In accordance with this approach, this research presents a forensic analysis of the non-volatile memory of the Windows 10 IoT Core operating system, since it is based on the most widely adopted OS in the history of computer science, and recent surveys suggest that it is the second most used IoT one [5]. Therefore, providing guidelines on how to examine it seems beneficial for the forensic community, as investigations in which this operating system is present will certainly be common in the near future.

### 1.2. Contributions

The contributions of this study are as follows:We study the current state of IoT forensics, explaining how the characteristics of the environment affect an investigation, and why traditional forensic techniques cannot provide a functional approach to be applied in this context.We present a review of the proposals from the community regarding IoT security and forensics.We conduct a forensic examination of the non-volatile memory of the Windows 10 IoT Core operating system. With this contribution, we address the study of a, forensically speaking, unknown operating system, thereby offering guidelines on how its analysis, acquisition, and evaluation should be carried out. This allows investigators to be able to rely on previous work when examining this system, easing the process.We list the relevant information that can be retrieved from the storage of Windows 10 IoT Core and which may be useful in a forensic investigation. This serves as a handbook to quickly observe what data can be extracted from the system and where it is located.We present a forensic tool to automatize the collection of these artifacts. This provides investigators with an IoT-specific program to properly study the non-volatile memory of Windows 10 IoT Core, instead of having to rely on general tools to perform this task.

The rest of the paper is organized as follows. Section 2 describes the Windows 10 IoT operating system, Section 3 discusses the related work in IoT security and forensics. An overview on how to perform a forensic analysis on Windows 10 IoT Core is provided in Section 4, and the pieces of evidence found in it are listed in Section 5. A tool to automatize their retrieval is proposed in Section 6. Finally, our conclusions are presented in Section 7.

## 2. Windows 10 IoT Core

Windows 10 IoT Core is the free version of the IoT-based operating system developed by Microsoft, namely Windows 10 IoT. It was launched in 2015 and it is a combination of the desktop and the mobile versions of the Windows 10 family, optimized for ARM and x64/86 devices such as Raspberry Pi, Dragon Board or Minnow Board [6]. The multi-language Universal Windows Platform (UWP) is the common app used to develop applications for the system, supporting C++, C#, JavaScript and Visual Basic. To remotely interact with the system, manage it and set it up, Microsoft provides the Windows 10 IoT Core Dashboard application, which can be installed on Windows 10 computers.

Some of the main features of this system are:Startup application, namely Windows 10 IoT Core Default App, to graphically interact with the system.Secure Boot: UEFI located security feature to only allow the execution of trusted applications signed by known authorities.Bitlocker encryption.Device Guard: allows the execution of only trusted code, identifying the firmware, drivers and applications that should run on the device [7].Cortana (no longer available since version 1809).PowerShell.Windows Update.Bluetooth.Web, SSH, and FTP Server.Compatibility with Arduino boards.Miracast.WiFi Direct.Other hardware compatibility such as WiFi Adapters, Ethernet Adapters, Cameras, NFC, RFID, and multiple sensors.

## 3. Related Work

In the following sections, the research carried out by the community regarding the security of the IoT and its forensic side is discussed. Although forensics is a subfield of security, a distinction is made to emphasize digital investigations performed on IoT devices, as this is what this work focuses on, but we also cover the current state of IoT security as it is necessary to understand its essentials before reviewing the forensics-related proposals.

### 3.1. IoT Security

The first security concerns arising from the IoT environment can be found in [8]. It presents several differences that the IoT architecture has, compared with the traditional ones, such as the formation of larger networks and the lack of a unified structure. The main resulting security problems are that data transmission is via wireless networks, meaning that signals are publicly exposed; the environment is very heterogeneous; and there are no universal standards for the development of IoT applications. This analysis is supported by [9], which also states that the approach for developing security mechanisms in the IoT has to be different from the one used in classical systems, due to the new features and characteristics of this new paradigm. In addition, a model based on nodes is proposed to represent the interaction between the main actors in the system and security practices. An interesting statement is made in [10] regarding the computational power of IoT devices, which causes, among other things, the need for a reduction in the computational requirements for cryptosystems or security applications, such as antivirus, in order to be able to use them. Supporting that idea, [11] adds that hardware-based security is the best approach for the IoT, and reviews the existing physical unclonable functions and their potential to be used as a security protocol.

A different perspective is offered by [12], in which IoT security concerns are classified according to the different layers that form the general IoT architecture, and a detailed analysis on how each layer should be protected is also presented. In addition, the most common threats for each layer are described, specifying what kind of attack they could individually suffer. The same standpoint is adopted by [13], but, instead of focusing on the security needs of each layer, it offers a more general perspective, evaluating security measures that affect all layers and reviewing the main ones taken by the community, which affect authentication, trust establishment, and security awareness.

Regarding the security solutions proposed by the community, in [14] a secure execution environment is developed in which a processing unit can execute applications in a protected manner, securing the device physically and not depending on a software solution that controls the processes in the system, adapting safety solutions to the characteristics of IoT devices and making the design of security systems a key task in the development process. Another interesting proposal is [15], which is focused on improving secure communications in IoT networks. A new routing protocol is introduced to authenticate devices when forming a network or joining an existing one, carrying out several tests to demonstrate that the security of the network has not been compromised by a malicious device, and that the overhead added by the protocol is almost insignificant. Ref. [16] tackles the problem of having unpatched and un-updated firmware on devices by developing a system that identifies the devices present in a network and makes a vulnerability assessment of each one of them. The communications established by every device are monitored and analyzed to decide whether they are a potential vulnerability or a harmless connection. For this purpose, a security gateway is used to monitor and control traffic and then, using a machine learning classification model, an evaluation is made to determine the isolation level required by a device, depending on the known common vulnerabilities and exploitations (CVEs) for it.

By focusing on the area in which IoT devices are used, we can also see how every context requires different security measures. In [17] a comprehensive analysis is performed, thoroughly studying two different IoT devices, namely a smart home sensor and an industrial smart meter. The security measures implemented on them were proven to be insufficient by carrying out several attacks that could cause a huge impact in a real scenario. Regarding the smart home sensor, they gained access to the root account of the device, its password, and the boot parameters, as well as being able to obtain the binary update file. Something similar occurred with the industrial meter, for which a modification of the ID of the device was successfully performed, which led to the possibility of making the device identify itself as if it were another. In relation to smart home security, ref. [18] presents the requirements that devices should meet in order to provide a trustworthy service, describing different components that can be found in the typical smart home infrastructure and highlighting, for each one of them, the security functions that they are supposed to provide.

### 3.2. IoT Forensics

A good starting point for understanding the current state of forensic research is [19], in which the problems that arise when using IoT devices are described. A key issue is highlighted, and that is the relationship between IoT devices and the cloud, which is an important feature when working in this environment. Another interesting article with a similar perspective is [20], which presents the different parameters of IoT forensics, such as the sources of evidence, the number of devices, the quantity and type of data, comparing this with traditional scenarios. In addition, two approaches are proposed on how to perform an IoT forensic analysis, stating the most relevant points to focus on. Data location and legal jurisdictions, as well as the difficulty of maintaining a chain of custody are some additional issues of the environment that are mentioned in [21]. To appreciate the wide range of IoT applications and what scenarios an investigator could face [22] is very useful; it also presents the taxonomy of IoT forensics as well as its requirements, offering a very complete analysis of the situation.

With these challenges in mind, solutions are proposed to facilitate analysis when dealing with IoT devices. One such solution can be found in [23], where a system is proposed to autonomously perform forensic tasks in an IoT environment, helping investigators to save time and automatize the analysis, allowing them to focus on obtaining information rather than spending time on trivial tasks such as parsing data, managing storage or creating timelines. In the quest for processing data more efficiently, the cloud emerges as an interesting possibility, as is stated in [24], which proposes a cloud-based service to perform forensic operations, allowing investigators to collaborate in an easier way and perform tasks more quickly, automatizing non-forensic actions such as resource management. Something similar is suggested in [25], where a model for performing IoT forensic investigations is designed, and guidelines are given to investigators on how to approach the analysis. Focusing on the identification phase, ref. [26] presents a highly detailed description of this process, extensively describing the phases into which it is divided, namely detection localization, recognition, and check-in. In addition, a selection method to provide the best evidence in a given scenario is proposed, based on the relevance, accessibility, localization and type of the data, illustrating the concept with a smart home device. Some tools are also proposed that take into account the characteristics of the environment, such as the one presented in [27], which allows the detection of duplicated digital images on digital media.

The immense diversity that characterizes the IoT environment leads to researchers focusing on studying specific devices. In [28] an investigation is carried out in order to determine what information stored in a smart TV can be important when performing a forensic analysis on it. In respect to smart TVs, in [29] the Amazon Fire TV stick is studied and guidelines on how to acquire a forensic image of the device when performing a chip-off are given, and a list is given of the artifacts that can be found on it, although the analysis is not very extensive. Other relevant devices are smart watches, which contain a considerable amount of sensitive information, as is shown in [30], in which two models are examined and a forensic analysis is performed on them, explaining the acquisition process and the tools that are used. The information obtained from them is not very relevant for an investigation, but the process followed is very interesting and significant in helping explain how to manage this kind of device. Due to the wide implementation of the IoT, we also find research regarding smart cities; in [31] recommendations are made on how to acquire and analyze the information that can be found in the electronic control unit of a car. Another vehicle-related study is [32], in which a useful term related to the IoT is introduced, namely the internet of vehicles (IoV). In this research, a framework is proposed for the recovery and storage of evidence that has been created in an environment that involves vehicles, networks, IoT devices, and cloud computing.

An exhaustive study of four different IoT devices is made in [33], following a six phase methodology. Information regarding the non-volatile memory, network, cloud and smartphone applications are acquired and analyzed. In addition, given the quantity of data collected and its structure, multiple plugins for the Autopsy tool were developed in order to extract information from it. The findings from each device are listed, and an interesting view is provided on how each phase can complement the others to overcome the challenges of the environment, such as accessibility or availability.

One of the major changes in digital forensics when dealing with IoT investigations is that the importance of the environment surrounding the device is far greater than in traditional analysis. The lack of computational process on IoT devices is balanced by the ability to exchange information with other similar systems, which greatly extends the range of forensic analysis. For this reason it is very useful to study an environment as a whole and not to focus only on examining devices individually. An interesting study is [34], in which an analysis of the Amazon Alexa ecosystem is made, examining the interaction of all the interconnected devices in that environment, such as mobile phones, computers, and smart speakers, and what data can be extracted from them and be used in a forensic analysis.

After studying the proposals of the community, it can be concluded that the study of specific devices and systems is a very popular and effective approach followed by researchers, which has produced several articles that have shed some light on how the forensic investigations in the IoT environment should be addressed. This, added to the information extracted from articles that were focused on studying the characteristics of the context, has created a solid base on which the community can work. On the other hand, there is little research centered on the creation of solutions for IoT forensics, although some small tools have been designed, but the frameworks and more complex services are still at very early stages of development, only offering a theoretical perspective. With this in mind, this article combines both types of proposals and introduces a forensic analysis of an IoT-based operating system, namely Windows 10 IoT Core, as well as presenting a small tool to be used in real investigations.

## 4. Analysis Method

This section presents how the forensic analysis of the non-volatile memory of the Windows 10 IoT Core operating system has been performed, describing in detail the components used, the methodology followed during the procedure and how it has been adapted to the characteristics of the experiment.

### 4.1. Test Environment

In order to carry out the analysis, it is necessary to establish and configure a proper environment to make sure that the experiment is performed correctly. In our case, the components used are the following:Raspberry Pi Model 3 B: host of the Windows 10 IoT operating system.32 Gigabyte microSD Card: non-volatile memory of the Raspberry Pi, as it does not include soldered storage.Windows IoT Core Build 17763: IoT version of Windows 10 released in February 2019.Desktop PC with Windows 10 and the Windows 10 IoT Core Dashboard application: acts as the forensic computer and it is also used to set up the Raspberry Pi and afterward connect to the device using the Windows 10 IoT Core Dashboard.Arduino board: used to test the connectivity of the system with other devices. To be specific, it is an Intel Galileo.Operative WiFi network: needed to study the effects of using a WiFi network on the device.

In Figure 1 a graphical representation of the environment can be seen, displaying how the Raspberry Pi interacts with the forensic computer and the Arduino Board.

### 4.2. Methodology

Even though the conventional forensic process models do not entirely suit the characteristics of the IoT environment, they can be adapted to the context in which this investigation takes place. In this case, the methodology is shaped by taking into account that the goal of the forensic analysis is to determine what information stored in the non-volatile memory of the system could be useful in a real investigation in which an incident has occurred. This translates into a more flexible process in terms of forensic soundness, since the examination is being carried out in a controlled environment that is specifically designed for the analysis, and the conclusions extracted from the analysis are not going to be used in a legal process. Therefore, certain measures such as the chain of custody are not required in this experiment. Obviously, the appropriate actions are carried out to avoid tampering with the data that is acquired and analyzed.

The process model used as a reference is presented in [35], in which an evaluation of the most relevant models produced from 1984 to 2011 is made, creating a generic one based on the commonly shared processes. The phases into which the methodology is divided are the following:Pre-process: preparation work that is executed before the start of the investigation, such as tool set up or warrant obtention.Acquisition and Preservation: refers to the identification, acquisition, collection, transportation, storage and preservation of the data.Analysis: study of the acquired data to extract information and draw conclusions.Presentation: documentation of the findings obtained and submission of the report to the authorities or the requester of the investigation.Post-process: relates to the closing of the investigation. Actions such as the return of the evidence are carried out in this phase.

In this analysis, the presentation and post-process phases are not necessary since the results of the investigation are not going to be presented in the form of a report, and the evidence acquired does not need to be returned. In addition, in order to adapt the model to the characteristics of the IoT environment, a new phase needs to be included in this analysis: evaluation. This refers to the procedure of grouping all the pieces of information collected from the different devices in the analysis phase and extracting conclusions from them about how they fit into the environment as a whole. In conventional process models, it is normally performed in the analysis phase, but, given the increase in the number of devices to analyze in IoT investigations, the task has become more complex and relevant, hence the creation of a new phase.

#### 4.2.1. Pre-Process

As no warrants or approvals are required to start the investigation, this phase consists of the design of the scenario in order to study the system and the tool preparation.

Scenario Creation. In order to be able to determine what information stored in the non-volatile memory is useful, it is necessary to acquire enough data to accurately capture the state of the operating system. To achieve this, three different scenarios that represent significant states of the operating system are analyzed. These scenarios, which help to understand the behavior of the system and the information that it handles, are the following:OS installation. This scenario allows us to study the system in its conception before any usage data is injected into it. The aim of this analysis is to comprehend how the data is distributed in the storage and to have a first contact with the operating system when it has not generated very much information. In addition, we examine what resources are used to configure the operating system in order to prepare for use. To create this scenario, after the microSD card has been sanitized, the “Windows 10 IoT Core Dashboard” program is launched, and the operating system is flashed into the storage. Once the installation process has finished, the microSD card is acquired and analyzed.First boot. In this case, we are trying to understand what information the operating system contains once it has been configured and is ready for the user to work with. Therefore, the system is studied when it is booted for the first time. All the terms are accepted, and the privacy settings are left at their default values. When the boot process has finished and the main screen is shown, the device is shut down and the non-volatile memory is acquired and analyzed.Normal usage. Lastly, the goal is to study the data generated by the operating system when the user has interacted with it. To achieve this, all the features of the system are explored: apps are installed and deployed, the settings are changed to fit the user preferences, a wired network and a wireless one are set up, connections with the IoT device are established using the Windows IoT Core Dashboard, services such as SSH and the web server are used, and the Arduino Board is paired via Bluetooth. After that, the storage is acquired and analyzed.

Consequently, three different analyses and acquisitions are performed during the experiment. The same procedure is followed in each one of them but, in some way, they can be seen as separate forensic examinations. Once all the scenarios have been independently analyzed, the results are put together and evaluated from a general perspective in the evaluation phase. The graphical representation of the methodology followed, combined with the scenarios studied, is shown in Figure 2.

Forensic Tools Used. Since at the time of this proposal there are not many forensic tools specifically designed for the IoT, general ones have been used to acquire and analyze the data stored in the non-volatile memory. Specifically, the selection of the tools was made on the basis of the knowledge that they are useful in retrieving evidence from multiple operating systems and, in particular, from the Windows 10 desktop version, on which the system that is being analyzed is based. The selected tools, which were all installed on the forensic computer, are the following:FTK imager: used for the acquisition of the non-volatile memory and for analysis purposes, since it has browsing and mounting capabilities [36].Autopsy: allows the investigator to browse through the storage, apply filters and recover deleted files [37].QPhotorec: data carving tool to recover the deleted files from a filesystem [38].Registry explorer: analysis tool that enables the browsing of the Windows registry [39].RegRipper: extract and interprets the data stored in the Windows registry hives [40].MFTExplorer: graphical viewer to display the content of the master file table (MFT) [39].AnalyzeMFT: parser to extract information from the MFT file in an NTFS filesystem [41].ESEDatabaseView: utility to read the data inside an extensible storage engine (ESE) database [42].

#### 4.2.2. Acquisition

Since the Raspberry Pi Model 3 B has a removable storage in the form of a microSD card, and it is physically accessible for the investigators, an offline acquisition is the best approach to follow. This process is carried out by executing the following actions:If the system is on, it is shut down. To do so, first, the system is turned off using the menu of the operating system, and then the Raspberry Pi is disconnected from the power supply. This guarantees that the storage does not suffer any damage or data loss.The microSD card is extracted from the board and inserted into a microSD to SD card adapter with write blockage capabilities.The adapter is then inserted into the forensic computer, making sure that the write blocker is on.The FTK Imager tool is launched and an image file of the storage is created.Once the image file is created, the hash value of the image is compared with that of the microSD card in order to guarantee that the data has not been altered.Finally, the image file is copied into a different storage location to ensure that at least one other copy is available in case the first one gets damaged.

The authors opted to create an image file of the storage instead of cloning it since it allows the analysis of the different scenarios simultaneously while consuming less physical resources. In addition, it also eases the preservation and storage of the data as the image files are saved on the forensic computer.

Regarding the preservation of the acquired image, it can be seen that no special measures are taken, just the essential ones necessary to certify that the data do not vary during the analysis of each scenario, thereby preserving the integrity of the evidence. In addition, every time that the image is mounted in the system, the hash value is calculated beforehand to assure that it has not been tampered with by an external element, and the read-only method is used.

#### 4.2.3. Analysis

To perform the analysis, the image file is mounted in the operating system using FTK Imager, selecting the read-only method, and then the pertinent tools are launched. For each of the scenarios, the actions carried out in this phase are:Analysis of the existing partitions in the storage, determining their purpose and what directories they contain.Examination of the directories of the different partitions to understand the operating system structure. These first two tasks are performed using Autopsy and FTK Imager.Study of the purpose of the different directories and what possible sources of evidence can be found in them. In this case, multiple tools are used to browse through the storage and to read the different file types that it contains.Carving of the deleted files in the filesystem to determine whether any relevant file has been removed. To do so, the QPhotorec tool is used.Comparison of the files stored in the microSD card between the different acquisitions, obtaining the hash value for each of the files to understand how the information varies on each partition depending on the actions that are executed on the system.

#### 4.2.4. Evaluation

In this experiment, although only one device is analyzed, the study of three different scenarios can resemble examining three distinct devices. Therefore, the evaluation phase is needed in order to establish which of the pieces of information retrieved from all of them is actually useful for consideration in a real investigation.

To do so, every piece of evidence or interesting piece of information that was found in the analysis phase of each scenario is studied to determine how it has varied during the experiment and how useful it is. For example, a directory that was relevant in the “OS installation” scenario may no longer be present in the “normal usage” one, so it has to be decided how plausible it is that an investigator can find it in a real investigation and how valuable it is from a forensic point of view. By following this approach, we make sure that only the most relevant data is selected and that all the possible sources of evidence are evaluated.

## 5. Forensic Evidence in Windows 10 IoT Core

Once all the different scenarios have been analyzed, and all the information gathered from them has been evaluated, the resulting data is the pieces of evidence that can be obtained from the system. In this section, this evidence or useful information that can be obtained from the storage is listed, detailing for every item the reasons why it could be useful in a forensic investigation. In addition, a summary of all the relevant artifacts found and their location is listed in Appendix A, which can be used as a handbook in future examinations.

### 5.1. Partitions

Knowledge of the distribution of the information over the different partitions of the system is essential to understand where the relevant data is stored, especially on IoT devices, which have limited storage space. As can be seen in Figure 3, three different partitions can be found in Windows 10 IoT Core: “EFIESP”, “MainOS” and “Data”. Their characteristics are the following:EFIESP: FAT 16 Extensible Firmware Interface system partition in charge of the booting process, in which boot loaders, applications and drivers that are launched by the UEFI firmware are stored. Its size is 63.7 Megabytes.MainOS: NTFS partition behaving as the system root directory that is launched by Windows Boot Loader when the device is turned on. Its size is 1.39 Gigabytes.Data: NTFS partition used by the system to store most of the information. It is the largest of all three available, and its size varies depending on the microSD card capacity since it takes all the space that is available after the creation of the “EFIESP” and “MainOS” partitions.

### 5.2. Directories

The files stored in the filesystems are cataloged using directories. Having a knowledge of what data they contain, and what they are used for, helps in finding evidence. In this case, the existing partitions have completely distinct purposes, so an in-depth analysis of the directories and files inside them was carried out to determine where the most relevant information can be found.

#### 5.2.1. EFIESP

Regarding evidence, this partition is not very relevant, considering that no information about system usage is found in it, as can be seen from its directory description presented in Table 1. In fact, most of the files did not vary between the different scenarios, thus maintaining the same hashes. However, an interesting file is stored in it when the operating system is burnt onto the microSD card: a provisioning batch file that calls another script located in the system drive that is used to configure the system with the preferences chosen in the setup process when it boots for the first time. In that script, the WiFi profile for the chosen network is created, and the password for the “administrator” user is set. Curiously, the data appears in plain text, so the WiFi key and the user password can be obtained. The file is deleted after the script is executed, but could be recovered by carving, compromising the security of the device and the network. Both files are shown in Figure 4 and Figure 5.

#### 5.2.2. MainOS

Due to its condition of system root, it is one of the key elements that have to be studied in this system. All the information that it contains is system-configuration-centered, meaning that there are a lot of useful files such as logs, registries and packages installed, but there is not much relevant data regarding the action of the users, as can be observed in Table 2.

#### 5.2.3. Data

This is the most important source of information if the investigator is focused on studying what actions have been performed by the users in the system, as it stores most of the user-related data such as applications installed, their data and the user registry hives. The directory structure of this partition is shown in Table 3.

### 5.3. NTFS Filesystem

Two of the partitions into which the storage is divided, and which are the most relevant ones forensically speaking, use the NTFS filesystem. Among its features, it contains multiple files that define and organize the filesystem, from which relevant information for an investigation can be recovered. In Table 4 a description of the purpose of each one of the files is provided.

In Figure 6, the content of the “$MFT” file of the “Data” partition is shown graphically using the MFTExplorer tool.

### 5.4. Registry

This is one of the main sources of information on Windows systems, containing data regarding users and system configurations, hardware devices and applications installed. The information is organized in a hive form, which is divided into registry keys, sub-keys, and values. The common registries that are normally present in a Windows desktop operating system are also available in the IoT version. In fact, they are present in the three existing partitions, although only the ones stored in “MainOS” provide useful information, the rest of them are almost empty. The most relevant registries of both the system and users are listed and described below.
System registry hives: they are located in Windows/System32/config.
–COMPONENTS: holds data associated with Windows Update configuration and status [45].–DEFAULT: profile for the Local System account. Used by programs and services that run as Local System such as winlogon or logonui [46].–DRIVERS: stores the drivers installed on the machine and their dependencies.–SAM: contains information used by the Security Accounts Manager. Among other data, it contains usernames and passwords.–SECURITY: collects local security information used by the system and network.–SOFTWARE: stores program variables and settings that apply to all the device users.–SYSTEM: contains device drivers and service configurations, which are stored in control set form [47,48].User registry hives: stored in the corresponding user directory and in Users\∗user∗\AppDate\Local\Microsoft\Windows.
–NTUSER.dat: stores personal files, preferences, and settings for each user [49].–Usrclass: used to record configuration information from user processes that do not have write permission to the standard registry hives. Information regarding shellbags is also stored here [50].

Some of the information that can be extracted from these registries is:Device details, such as number of cores, amount of storage and memory.Partitions on the system.Location of the Default Application, Temp, Program Files and Common Files paths, among others.Packages installed in the system.Digital certificates.Network profiles.Bluetooth devices paired.Mounted devices.USBs connected.Drivers installed.Browser history and settings.

Another relevant registry file present in the system is “Amcache”, which stores information about the executed applications. As can be seen in Figure 7, data such as the executable name and location, its hash, its version or the program identification number can be found for the SSH service. It is stored in the following route: Windows\AppCompat\Programs\Amcache.hve.

### 5.5. System Events

These contain the logs of all the relevant actions occurred in the operating system classified into four different categories, depending on what component of the system was affected:Application: activities regarding the software and components installed on the system.Security: data regarding the Windows system audit policies.Setup: data about the control of domains.System: mainly events related to the Windows system files [51,52].

They are also categorized according to the impact that the action has had on the system in errors, warnings or information messages, ordered from least to most relevant. For these reasons, the event log is a very useful source of evidence, added to the fact that the information is presented in a very detailed and structured way, making it easy to filter through the large amount of data that it stores. The logs for each category can be found in the following route of the MainOS partition: Windows/System32/winevt/Logs/.

Other relevant events that are also stored are the ones regarding the following services:OpenSSH: information about the log attempts, launch, and stop of the SSH server can be recovered.Network profile: data regarding the network and the connection type of the system, as well as information of when the system disconnected from it.Windows update: events are created when an update is found or downloaded.AppXDeploymentServer: information regarding the packages that have been installed and uninstalled on the system.

In Figure 8, an example of a system event related to the Network Profile is shown, in which the disconnection of the system from the WiFi network is logged.

### 5.6. Users

The actions that occur in a system can have a different impact depending on who executed them. For this reason, it is very useful to be sure of what permissions every user in the system has, and what their purpose is, as this facilitates the task of understanding how the changes that a system has undergone could have been made. In Windows 10 IoT Core the following users are present in the system:DefaultAccount: system managed account, member of the System Managed Accounts Group. This is the account used to log into the system every time that the operating system boots.DevToolsUser: account used to develop UWP applications.System: administrator account used by the system with maximum privileges to access all the data.Administrator: account for administering the computer protected by password, and set during the SD creation process. This is the user required by the Windows 10 IoT Core Dashboard application in order to connect to the system.Guest: restricted account for guests to access the system. Disabled by default.WDAGUtilityAccount: disabled account managed and used by the system for Windows Defender Application Guard scenarios.sshd: non-privileged account that has no info about its purpose but, as its name shows, it is used by the system to manage the OpenSSH service.

Regarding the data that is stored for every user, as can be seen in Figure 9, their directories have a similar structure to that of the desktop version of Windows 10, also maintaining the “AppData” folder to store information regarding an application’s personal configuration, which is divided into “Local”, “LocalLow” or “Roaming”, depending on whether the settings are stored in that device only (“Local” and “LocalLow”) or synchronized with others (“Roaming”).

### 5.7. Apps

Similarly to smartphones, the programs installed in Windows 10 IoT Core are present in the form of apps. Since they are the ones that provide meaning to a system, their relevance in a forensic analysis is very high, firstly because of the useful data that they store, and, secondly, as they help to understand what the purpose of the device is.

The apps installed on the system are stored in the Programs\WindowsApp route of the “Data” partition, in which the user configuration data is stored. The process involved in an app installation is the following:A directory is created for the app in \Programs\WindowsApp, where it will be installed.The packages needed for the app are stored in \ProgramData\Microsoft\Windows\AppRepository\Packages.The user information of the app is saved in their local directory: \Users\DefaultAccount\AppDate\Local\Packages\.

Thus, in conclusion, apps behave as a program does in the desktop version: they are installed in a directory, then the general information is stored in a common folder so it can be accessed by any user that launches the app and, finally, each user has a folder created in their local directory where the configuration and program data is stored.

In Figure 10, the \Programs\WindowsApp directory is shown, where it can be seen that three applications that were not originally on the system have been installed.

### 5.8. Browser

Although IoT devices are not designed to be used for web browsing due to their computational capacities, this operating system provides a browser. Therefore, it must be analyzed, as it is one of the mandatory sources of evidence in a forensic investigation, especially when studying a desktop system or a smartphone, in which the relevance of this data is much greater. After the registry analysis, information with respect to the user agent of the native Windows 10 IoT browser was found, determining that it is “Mozilla/5.0 (compatible; MSIE 9.0; Win32)”, an outdated and vulnerable version. Also, data regarding the web pages visited, cookies and cache can be extracted from the registry and the app folder for the browser, specifically from the following locations:Users\DefaultAccount\AppData\Local\Microsoft\Windows\WebCache.Users\DefaultAccount\AppData\Local\Microsoft\Windows\INetCache.Users\DefaultAccount\AppData\Local\Microsoft\Windows\INetCookies.

An example is presented in Figure 11, in which the browsing history is extracted from the WebCache file.

### 5.9. Bluetooth and WiFi Connections

Taking into account the importance of connectivity in this environment, Bluetooth and WiFi data are among the most relevant that can be found in an IoT system. Almost all IoT devices are compatible with these technologies, especially WiFi. The best location to look for such evidence is the registry. Regarding WiFi, data such as interface configuration, network interface cards available or wireless profile settings can be extracted from the “SOFTWARE” registry file in SOFTWARE\Microsoft\WindowsNT\CurrentVersion\NetworkList, as shown in Figure 12. In addition, a WLAN event log file, in which data of the wireless networks associated with the system is stored, is also available.

In the case of Bluetooth data, the IDs and names for the devices connected are stored in the registry “SYSTEM” in SYSTEM\ControlSet001\services\BTHPORT\Parameters\Devices. In Figure 13, the result of interpreting that key with RegRipper can be seen.

### 5.10. Pagefile and Hiberfil

The exchange of data between the physical memory and the persistent storage leaves very useful sources of forensic information, such as the hiberfil and pagefile files, both of them available in Windows 10 IoT Core. They are stored in the root directory of the “MainOS” partition. In order to be created, the option has to be enabled in the system registry, which is not the case of the hiberfil file, as the hibernation option is not active. They contain the following information:Hiberfil: file used to save the state of the device when the system is put in hibernation mode. It contains data that, instead of being stored in RAM memory, is saved temporarily in the storage before shutting down the system, and then recovered when it restarts, such as user passwords, deleted files, connections established or information about running processes, among other data. As can be seen in Figure 14, after enabling the hibernation mode and putting the system in that state, the file is created.Pagefile: contains data temporarily exchanged between RAM memory and persistent storage. This occurs when the system needs more space available in physical memory, so virtual memory is created through paging, therefore storing a piece of information about RAM memory in the pagefile file. In order to analyze it, a tool such as Volatility [53] must be used to interpret the file content.

## 6. Proposed Tool for Evidence Retrieval from the Windows 10 IoT Core Non-volatile Memory

In this section, a target for the forensic tool Kroll Artifact Parser and Extractor (KAPE) [54] is developed to collect the relevant sources of information that have been found during the analysis process of the non-volatile memory of Windows 10 IoT Core.

### 6.1. KAPE

KAPE is a program developed by Eric Zimmerman that allows investigators to collect forensically useful artifacts from an evidence source file, which can be present in the form of a live system or a mounted image, and process them using well-known forensic tools, making the collection and analysis processes in an investigation considerably more effective and quicker. Its functioning is based on modules and targets, which can be easily programmed to provide new functionalities to the tool. Targets are used to recover the relevant files and directories from the source file, and modules are in charge of running the forensic program that is capable of interpreting the relevant file and extracting information from it. Both of them are written using YAML, and can be executed on a Windows system using the command prompt, PowerShell or via its graphical interface. Some of the features provided by it are the following:Targets: some examples of what data can be recovered by the tool are:
–Evidence of execution, shortcut files and jump lists.–Metadata of the filesystem.–Antivirus logs.–User files.–Scheduled tasks.–Web browser data, such as history, bookmarks or cookies.–USB device log files.Modules: using the multiple forensic tools included in KAPE, these actions, among others, can be performed:
–Event log parsing.–Registry information extraction.–Timeline creation.–File accessed listing.–Prefetch files processing.–Browsing history access.–Extraction of program execution data.

### 6.2. Target Developed for Windows 10 IoT Core

Since there are not many IoT forensic tools, and none of them are compatible with Windows 10 IoT Core, a target is programmed in KAPE to facilitate the evidence retrieval process when investigators examine the aforementioned operating system (the target developed can be downloaded from the following link: https://bitbucket.org/juanmanuelcastelo/windows-10-iot-collection-target/src/master/). The artifacts to collect are the ones described in Section 5 and listed in Appendix A, which have been confirmed to be relevant after the forensic analysis performed. A piece of the code programmed is shown in Figure 15.

To properly recover the data, the target has been divided into two files, one for the “MainOS” partition and another for “Data”, since they have some homonymous directories and using only one target would lead to the collection of non-relevant artifacts. As the program only allows one target source, it has to be executed twice, once for each partition. The result of executing the target developed can be appreciated in Figure 16 (the –tflush must be used in the first execution, but must be omitted in the second, or the target directory will be deleted).

As can be seen, the total time employed for the collection of 401 artifacts was 53 s, which is considerably faster for an investigator than having to browse through the directories and extract the files one by one. Therefore, instead of the process taking days, which was the case in our experiment, it can be performed in only seconds. In addition, it also automatizes the task, allowing the examiner to be able to pay more attention to the analysis phase, rather than focusing on obtaining the possible sources of evidence in the system, which, when one knows which and where they are, is a trivial operation that requires too much time and delays the investigation.

## 7. Conclusions

In this research, the reasons for the recent increase in the number of cybersecurity incidents involving IoT devices and systems have been detailed. The weaknesses in their security measures have been discussed, highlighting the need to improve them, given the sensitivity of the information that they handle, added to the important role they have in certain contexts, such as critical environments. It is essential that cyber criminals should not find it easy to compromise them.

Regarding IoT forensics, it has been explained why the conventional forensic approach cannot satisfy the requirements of the IoT environment, so there needs to be an improvement in the field in order to provide techniques to perform complete and efficient investigations. In addition, the characteristics of IoT devices have been described in order to comprehend what features make this context unique and how they affect the examinations. On reviewing the related work, it has been recognized that an effective method to address this issue is by analyzing IoT devices and systems in order to understand how they operate and what information can be retrieved from them.

By following this approach, a forensic analysis of the non-volatile memory of the Windows 10 IoT Core operating system has been performed, offering guidelines on how to conduct it, detailing aspects such as the analysis, acquisition, and evaluation of the pieces of evidence detected. This provides investigators with a study that they can use as an aid when investigating the same operating system, especially when, at the time of this proposal, there are no specific IoT methodologies to follow.

Furthermore, the sources of relevant information that can be retrieved from the storage have been listed, creating a useful handbook that describes what artifacts have been identified, their purpose and location. In addition, it has been seen that the desktop Windows 10 version and the IoT-based one share relevant characteristics and data, consequently meaning that studying other similar systems prior to performing an investigation can very beneficial in order to determine how to approach it, particularly when they are based on the same concept.

In addition, it has been proven that the forensic examination of IoT systems ultimately leads to the development of specific tools that can facilitate the investigation process, making it more efficient than when only general ones are used. A module for KAPE, a forensic program, has been developed to collect all the relevant sources of information stored in the non-volatile memory of Windows 10 IoT Core. This allows investigators to automatize the evidence retrieval task in future investigations in which this operating system is present. Furthermore, it also shows that the increase in functionality in general forensic programs is a useful approach to follow when developing IoT tools, instead of focusing on creating independent ones.

### Future Work

This work has been an introduction to IoT forensics, in which the need for an improvement in guidelines, techniques, methodologies, and tools available for investigators has been made evident. Therefore, there is a wide spectrum of research that needs to be carried out in order to ensure that examinations are performed in a complete and efficient way. Some of these projects could be the following:Extend the analysis of the Windows 10 IoT Core operating system, examining it from a dynamic perspective, focusing on studying the volatile memory and network traffic in order to have a complete understanding of what evidence is contained in it and how to collect it.Perform further research to study the similarities and differences among the different operating systems of the Windows 10 family, with the aim of discovering new possible evidence or techniques that can be applied in an IoT context.Continue the development of forensic tools to allow investigators to automatize the examination process, making them more efficient and simpler.Provide guidelines on how to analyze other IoT-based devices or systems, so that analysts can make use of other research when having to study them.Enlarge the scope of investigations with the goal of understanding the interaction between IoT devices and systems from a forensic viewpoint since connectivity is the main feature of this environment.

## Figures and Tables

**Figure 1 entropy-21-01141-f001:**
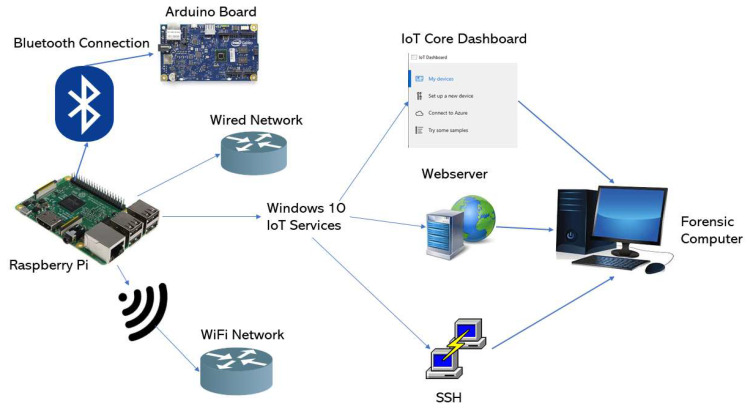
Test environment established.

**Figure 2 entropy-21-01141-f002:**
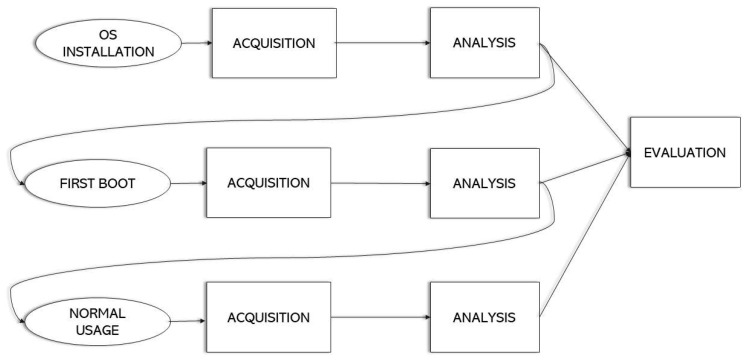
Methodology followed to perform the forensic analysis.

**Figure 3 entropy-21-01141-f003:**
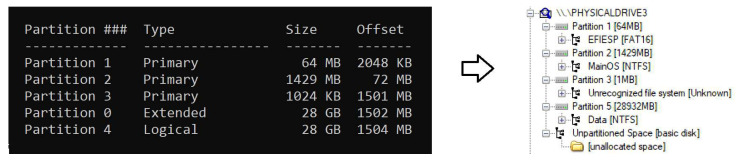
Partitions into which the storage is divided into.

**Figure 4 entropy-21-01141-f004:**
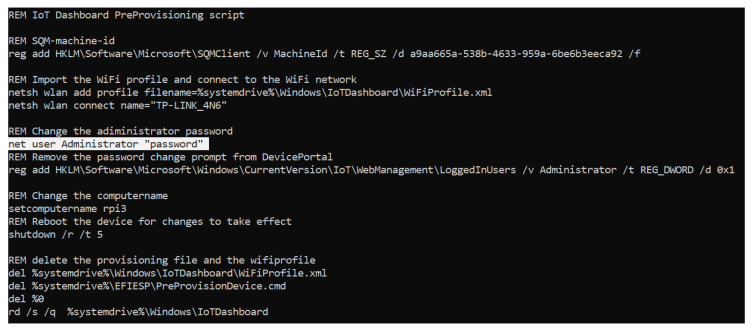
Batch pre-provision file for the system set up.

**Figure 5 entropy-21-01141-f005:**
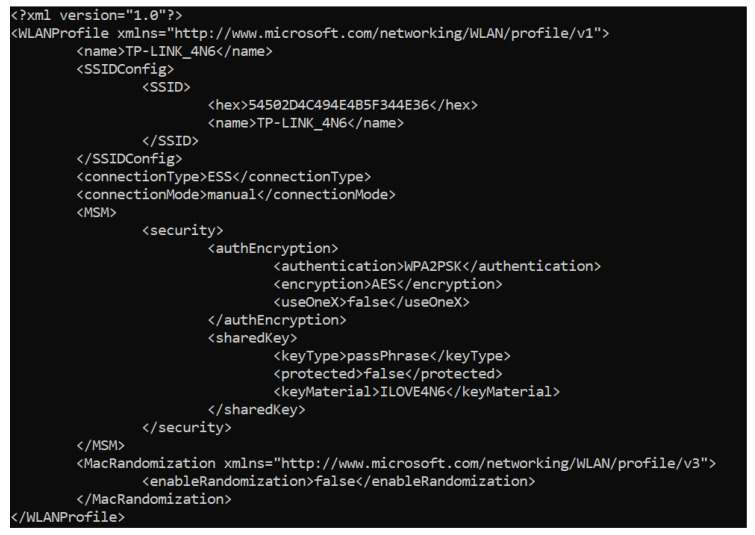
WiFi profile file created when the microSD is burnt.

**Figure 6 entropy-21-01141-f006:**
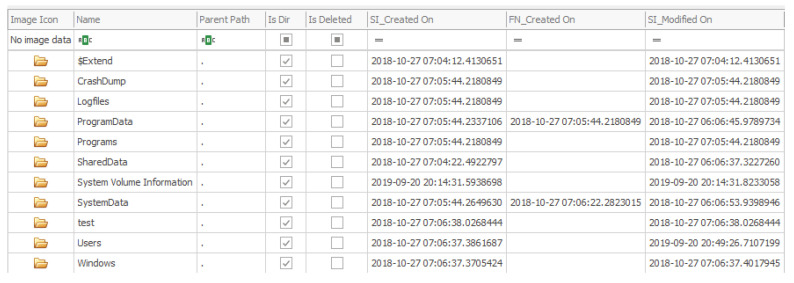
Graphical view of $MFT file content.

**Figure 7 entropy-21-01141-f007:**
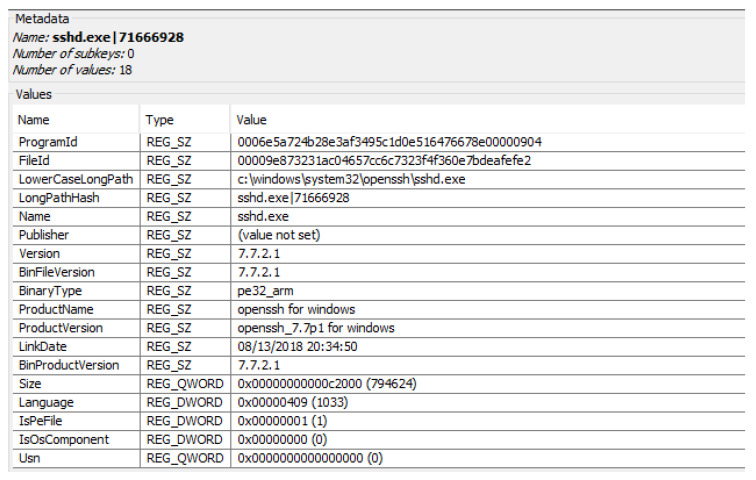
Value of a subkey of the Amchache registry file.

**Figure 8 entropy-21-01141-f008:**
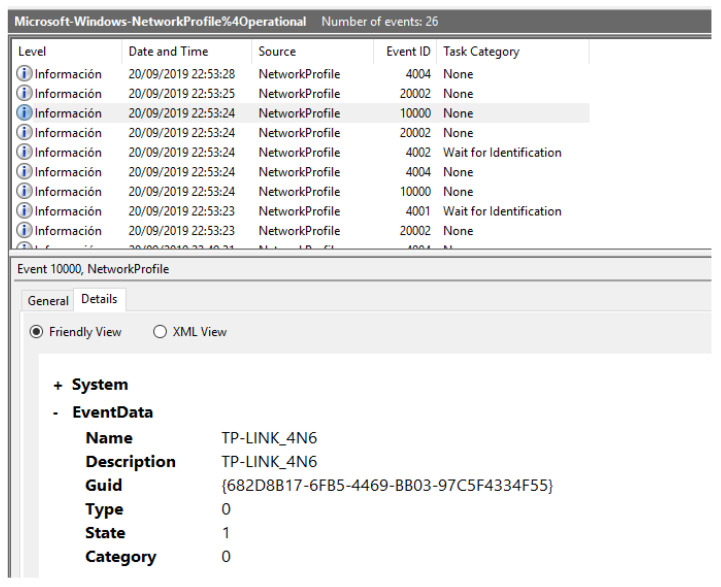
System event informing of the disconnection of the system from the WiFi network.

**Figure 9 entropy-21-01141-f009:**
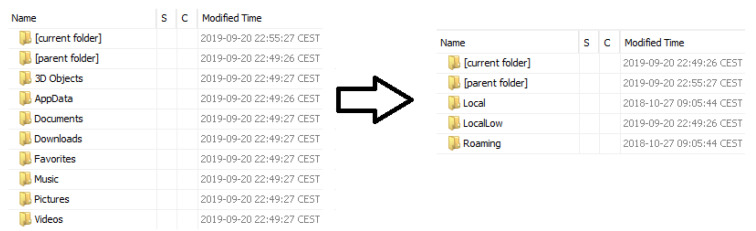
Content of the user directory and “AppData”.

**Figure 10 entropy-21-01141-f010:**
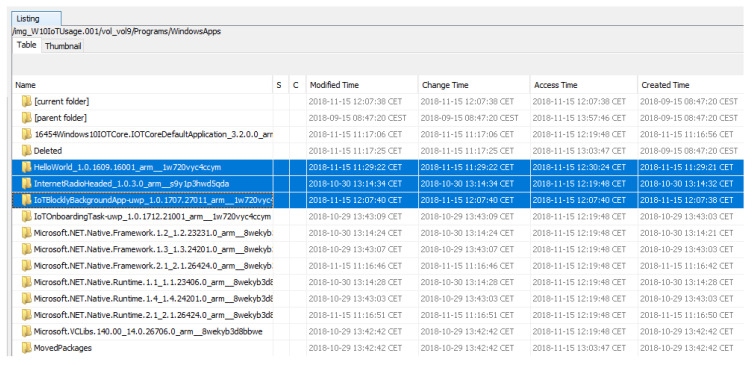
Apps installed on the system.

**Figure 11 entropy-21-01141-f011:**
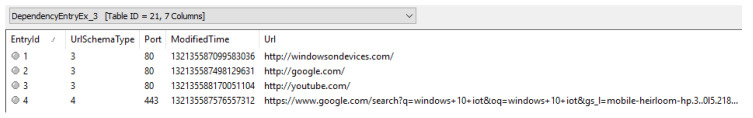
Websites visited using the native Windows 10 IoT Core browser.

**Figure 12 entropy-21-01141-f012:**
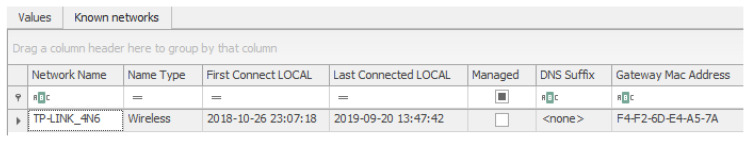
Entry in the registry for the known WiFi networks.

**Figure 13 entropy-21-01141-f013:**
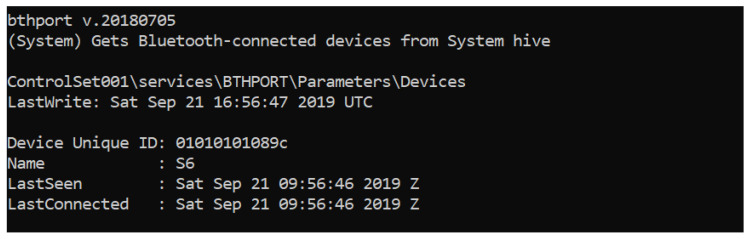
Registry entry logging the Bluetooth devices paired.

**Figure 14 entropy-21-01141-f014:**
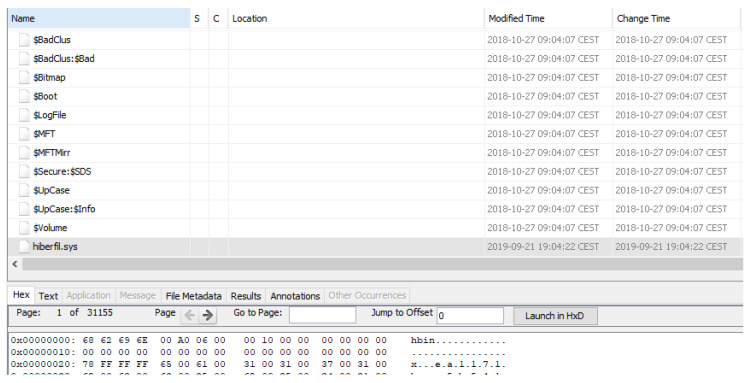
Hiberfil file stored in the MainOS partition when the system is hibernated.

**Figure 15 entropy-21-01141-f015:**
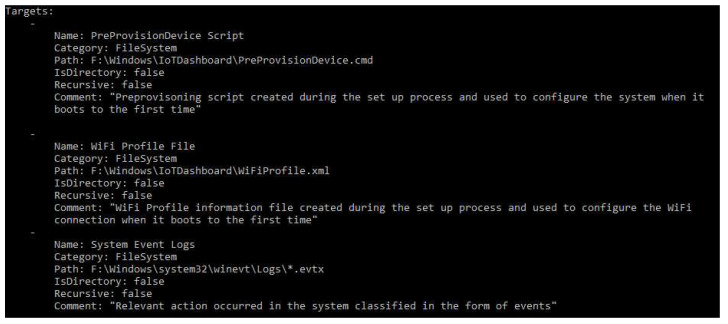
Piece of the programmed target for evidence collection in Windows 10 IoT Core.

**Figure 16 entropy-21-01141-f016:**
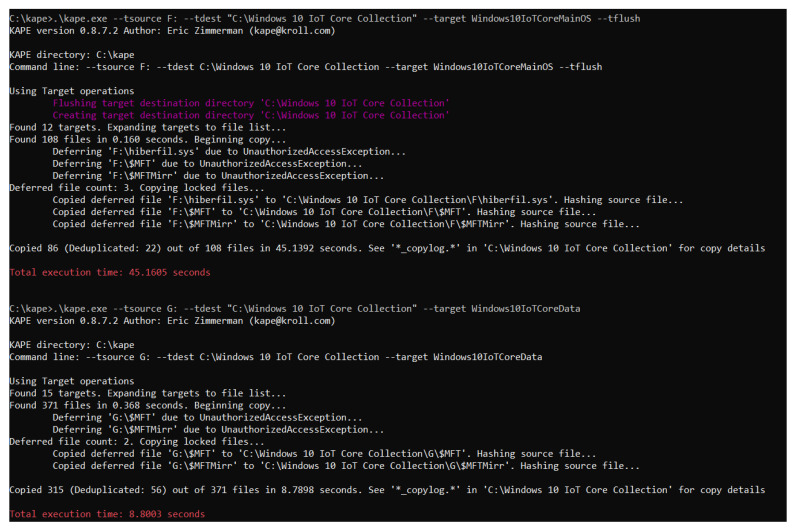
Execution of the target developed.

**Table 1 entropy-21-01141-t001:** Directories in the EFIESP partition.

Directory	Description
boot	Installation boot files.
EFI	Contains the booting information and settings for the operating system startup process. We can find the bootloader for the ARM architecture and the boot configuration data.
System Volume Information	Folder used by the system to store its information and the restore points.
Users	Directory in which information about the system’s users is stored. Here we can find that a user account under the name “default” exists, but no relevant material is found in it, including the NTUSER registry. This user profile is used as a template when creating new users, which will be built based on the “default” profile.
Windows	Typical Windows system structure directory. Information about specific IoT and ARM packages can be found, but nothing relevant as no usage information is stored in this partition.

**Table 2 entropy-21-01141-t002:** Directories in the MainOS partition.

Directory	Description
$Extend	Folder that contains metadata and optional extensions regarding the NTFS filesystem.
Data	It is a symbolic link to the Data partition, known in NTFS as a junction point.
Program Files	Directory to store the information of the programs that are installed on the system. This folder is not used for this purpose in this operating system, as the programs are installed on the “Data” partition.
ProgramData	Folder used for application data that is not user-specific, that is to say, the information is available for every account in the system, so all the information of the programs that are shared between the users is stored in this directory. It has no forensic value as the “ProgramData” directory on the “data” partition stores most of the information.
PROGRAMS	Folder typically used on Windows 10 Mobile to store the preloaded apps in the system, although there is no information stored in it in the IoT operating system. We can find the folder used to update the system,“UpdateOS”.
System Volume Information	Folder used by the system to store its information and the restore points.
SystemData	Contains a directory named “Temp” with no information in it.
Users	Local information of the users can be found here. There is a “default” user used for the start menu and tiles design, and a “public” directory for the namesake user, but they are empty.
Windows	As this partition behaves as the system directory, the Windows folder contains information about the packages installed, drivers, executables, libraries and relevant forensic artifacts such as the system registry hives and the system event logs, among other data. There is also a directory in “System32” named “LogFiles” in which a log about the connections made to the webserver can be found.

**Table 3 entropy-21-01141-t003:** Directories in the data partition.

Directory	Description
$Extend	Folder that contains metadata and optional extensions regarding the NTFS filesystem.
CrashDump	Keeps the information stored in memory when the system or an application crashes.
Logfiles	Directory in which the logs from different applications are stored. Complements the namesake directory that is available in the “MainOS” partition, although in this analysis only information about the Windows Management Instrumentation (WMI) has been found.
ProgramData	Same purpose as the directory in the system root partition. In this folder there is more information than in the aforementioned one, and the data regarding the SSH service is especially relevant. Also, the packages that have been installed for the different applications in the system can be found here.
Programs	Contains directories for the different applications that have been installed on the system as well as a folder with the deleted ones. Each folder contains the resources needed for that application to run.
SharedData	Folder designed for shared storage.
System Volume Information	Folder used by the system to store its information and the restore points.
SystemData	Contains a directory for the Event Tracing for Windows (ETW) logs, a different one for the non-ETW ones and a “Temp” folder.
test	Used for the Windows Driver Test Framework (WDTF) for developing and running tests on the system.
Users	Local information for the users of the system are stored in this directory. The most relevant user is the “administrator” as this is the one that is logged on automatically when the system boots, therefore being the one who executes all the actions that are performed by a user on the device.
Windows	Little data can be found in this directory, as the relevant “Windows” folder is the one stored in the “MainOS” partition. Some packages are stored in this partition as well as the system registry files, which are almost empty.

**Table 4 entropy-21-01141-t004:** Metadata files in NTFS.

File	Description
$AttrDef	Describes the attributes supported on the volume. It is essential for the filesystem, since a file is a representation of these attributes.
$BadClus	Informs of the clusters that contain bad sectors.
$Bitmap	Contains the status of the clusters in the filesystem.
$Boot	Provides data with respect to the booting process such as the boot sector.
$I30	Representation of the $INDEX_ROOT, $INDEX_ALLOCATION and $BITMAP attributes. They present information regarding the filenames and directories stored in a specific directory.
$LogFile	Stores the transactions that have been performed in the system to allow their recovery after a failure.
$MFT	Most important file of all, as it is a table that contains information for every file and directory that has been stored in the filesystem. It is extremely useful in forensic investigations as it logs all the activities that have occurred, providing information regarding timestamps, attributes, names or how it was created, among other data.
$MFTMirr	File to allow the recovery of the MFT.
$Secure	Lists the security descriptors for the volume.
$TFX_DATA	Contains transactional data. Corresponds to the $LOGGED_UTILITY_STREAM, attribute, but some tools such as FTK Imager represents it as an independent file [43].
$UpCase	Used to compare and sort filenames.
$Volume	Describes the characteristics of the volume, such as its identifier, label or version [44].

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
