# Peer review of "Non-Volatile Memory Forensic Analysis in Windows 10 IoT Core"

_entropy, 2019, doi:10.3390/e21121141_

Round 1
Reviewer 1 Report
In this article, the state of IoT forensics is studied, also reviewing the security needs of an IoT system. In addition, a forensic analysis of the non-volatile memory of an IoT operating system, 10 namely Windows 10 IoT Core, is presented, explaining how to perform it and highlighting the most 11 relevant information that can be found in it. The authors have put an effort in trying to address a number of aspects that are related to IoT OS from Windows, but there seem to have ignored very critical aspects. Currently, the way the paper, I believe it needs a lot of improvements before it can be considered for publication. Firstly, the motivation of the study is not so critical and not entirely convincing hence the novelty of the study comes into question. The grammar in this paper is poorly done, this paper must be edited professionally by language editors. The abstract alone is really hard to understand. Also, the relationship between research in the related work has not been correlated well, more or some of the concepts can only be viewed from far, indicating that some aspects are not really needed. Another key concern is the methodology-it is not clear what methodology has been employed in this research, I believe given the mixed approaches, it is hard for the reader to easily follow what happens in each step because of lack of mention. The authors need to mention what methodology they have employed in this research. Normally, a digital forensic process has steps which sometimes are systematic and sequential-the study has mainly chosen to ignore that aspect. In the analysis in Section 5, no forensic approach can or could be seen, the author needs to document systematic digital forensic process, together with the technical aspect. Mention for example, where inception starts, when you do acquisition, how examination, preservation, transportation, how the process is secured, presentation and reporting etc. also, the literature is too much and some things are really not needed or not relevant to the study. Choose a forensic process model and tell the reader how that has been broken down in this study. While the study seems to be focused more on the technical aspects, it is also important to highlight the demarcation of IoT security and forensics. In section 6, the author has a mention useful information, the question I would like to ask, in the context of this research, what is considered useful and what is considered not useful. While it is true that the paper uses some existing tools (line 335-344) to achieve the desired objectives, it becomes hard to identify where or how each of the processes are achieved. Let the authors dissect the paper and fit to what translate to each forensic process and also, the authors need to address how forensic soundness and the sanctity of the collected or accumulated digital evidence can be guaranteed. Otherwise, the way the paper is, one could argue that it ignores vital/key aspects that are very important. I would like to see that some of these concerns addressed before this paper can be accepted.
Author Response
Dear Sir/Madam,
We would like to thank you for your valuable comments, which have made it possible to significantly improve the quality of the paper. We have made some changes to the manuscript, trying to address all the comments received. Please find attached a document describing how each of the comments have been addressed, as well as a description of other significant changes.
Sincerely,
Juan Manuel Castelo

Reviewer 2 Report
The topic is very timely, needed, and hot. It covers a clear gap, and there is a lot of research going on currently in this area. Therefore, this paper deems an excellent future reference for researchers who work in the area. The overall quality is good and meeting the minimum requirements of the journal. The content is relevant and useful for the researchers’ community. However, it has many minor issues that must be fixed properly ranging from literature and lack of suitable directions etc as follows:
The abstract is long and doesn't communicate the problem well. And, it should present the proposed work more clearly. The abstract must summarise the performance evaluation results and improvement over competitors and/or other solutions. Please make sure that all keywords have been used in the abstract and the title.
Introduction section is not clear. In other words, it is not introducing the contributions, motivations and the problem clearly. I suggest you create subsections in the introduction section to better organise it. One of the new subsections must be for the problem discussed and one for the contributions. The novelty of the proposed work should be clearly introduced. Please use bullet points for listing the contributions.
The related work is scattered throughout the text and doesn’t cover the literature and state-of-the-art of the presented work appropriately. Therefore, I strongly recommend that the authors add more literature about this point. Some very relevant and recent work that the authors may use to address this point: https://ieeexplore.ieee.org/abstract/document/8328748; https://www.sciencedirect.com/science/article/pii/S1742287619300222 https://ieeexplore.ieee.org/abstract/document/7792460; https://dl.acm.org/citation.cfm?id=3233257;
Results should be further analyzed.
The conclusions section should conclude that you have achieved from the study, contributions of the study to academics and practices, and recommendations of future works.
Thorough proofreading is required (best by a native English speaker)
Author Response

(The authors gave the same response as above.)

Round 2
Reviewer 1 Report
The authors have taken care in addressing the comments. The paper can be accepted in its current form but the following could be addressed.
According to a Gartner estimation [1], in 2018 there were more than 11 thousand million IoT 22 devices installed, and an increase of almost twice this value is predicted for 2020, with 20.4 thousand millionunits. -In line 21, the authors can choose to talk of billion devices instead of thousand million, kindly find the equivalent of the figure if 20.4 billion or what the figure by Gartner looks like.
Line 29-Kindly use IoT instead of Internet of Things, because you have already introduced the acronym. Apply this throughout this paper.
Line 46-Put the DDoS abbreviation after the expansion not before-Apply throughout the paper.
In section 1.1: A full colon can be inserted after the following that looks like subtopics, purpose, connectivity, computational capacity, location and heterogenity.
Reviewer 2 Report
I would like to thank the authors for addressing all comments sufficiently and appropriately. The paper has been improved significantly, and I have no more comments. I recommend accepting the paper for publication in the present form.